# Effect of Deposition Aids Tank-Mixed with Herbicides on Cotton and Soybean Canopy Deposition and Spray Droplet Parameters

Chase Allen Samples [1], Thomas R. Butts [2], Bruno C. Vieira [3], Jon Trenton Irby [1], Daniel B. Reynolds [1], Angus Catchot [4], Greg R. Kruger [3] and Darrin M. Dodds [1,*]

1   Department of Plant and Soil Sciences, Mississippi State University, Mississippi State, MS 39762, USA; CSamples@americot.com (C.A.S.); trent.irby@msstate.edu (J.T.I.); daniel.reynolds@msstate.edu (D.B.R.)
2   Department of Crop, Soil, and Environmental Sciences, University of Arkansas, Lonoke, AR 72086, USA; tbutts@uaex.edu
3   West Central Research and Extension Center, University of Nebraska-Lincoln, North Platte, NE 69101, USA; bruno.vieira@unl.edu (B.C.V.); greg.kruger@unl.edu (G.R.K.)
4   Department of Biochemistry, Molecular Biology, Entomology and Plant Pathology, Mississippi State University, Mississippi State, MS 39762, USA; alc4@msstate.edu
*   Correspondence: dmd76@msstate.edu

**Abstract:** The adoption of auxin-tolerant crops has increased awareness regarding herbicide off-target movement. Deposition aids are promoted as a possible solution to off-target movement, although their effect on spray canopy deposition are not well understood. Studies were conducted to determine the impact of deposition aids tank-mixed with herbicides on spray droplet size and canopy deposition. Commonly used herbicides were applied on soybean and cotton in combination with deposition aids (oil, polymer, and guargum). Interactions between herbicide solution and deposition aid influenced droplet size parameters for both cotton and soybean herbicides tested herein ($p \leq 0.0001$). Generally, the addition of polymer and guargum deposition aids increased spray droplet size, whereas the addition of oil deposition aid decreased droplet size for some treatments. When herbicides were combined, the inclusion of deposition aids did not influence overall spray deposition on cotton ($p = 0.82$) and soybean ($p = 0.72$). When herbicide solutions were evaluated individually, the advent of deposition aids had inconsistent results with cotton and soybean spray deposition being unaffected, increased, or even decreased depending on the herbicide solution tested. For example, the polymer-based deposition aid increased spray deposition on cotton for applications of glyphosate + dicamba + S-metolachlor resulting in 1640.6 RFU (relative fluorescence units). However, the same deposition aid decreased spray deposition on cotton for applications of glyphosate + dicamba + acetochlor (1179.3 RFU). Although deposition aids influenced spray deposition on cotton and soybean for some herbicide combinations, their use should be determined on a case-by-case scenario.

**Keywords:** adjuvants; dicamba; 2,4-D; glyphosate; glufosinate; herbicide-tolerant crops; spray droplet size





## 1. Introduction

The application of pesticides is a complex mechanical process that results in a complex biological response [1]. Advances in pesticide application technology with nozzles designs [2,3], pesticide formulations and adjuvants [4,5], spraying techniques [6], and strategies to mitigate spray drift [7] are essential to ensure satisfactory pest control while mitigating the negative effects of pesticides to the surrounding environment. Droplet size and velocity distribution are two of the most important factors in accuracy and retention of pesticide applications [8]. The shift to glyphosate-tolerant crops intensified the need to control off-target movement of glyphosate onto surrounding non-resistant plants [9,10].

Numerous cases of legal action involving off-target movement of pesticides are filed with the Mississippi Bureau of Plant Industry each year. Of these, a large majority involve herbicides. With auxin tolerant crops gaining regulatory approval, off-target movement of auxin herbicides has been deleterious for adjacent crops and other species susceptible to these herbicides. Droplet size is manipulated by nozzle selection, application pressure, and nozzle orifice size [3]. In addition to this, off-target movement of spray particles may be mitigated by increasing the droplet size with a drift control adjuvant in combination with the pesticide [11].

Adjuvants are grouped into categories based on the effect they have during the application process, function, and the chemical class to which they belong. Some products are multi-use adjuvants which are typically the result of specific physiochemical properties of the adjuvant [12]. Utility adjuvants may influence spray formation which becomes important when applications require an optimum droplet size for activity [13]. However, utility adjuvants generally do not affect herbicide efficacy but rather attempt to make the application process more efficient [14].

Adjuvants used to control off-target movement are commonly referred to as drift control agents (DCA). There are several types of DCAs with the most common being polymer or polymeric products [15]. A major group of polymeric DCAs is polyacrylamide-based products [14]. Among this group of DCAs are polysaccharides, with the most common being guargum and xanthan gums [14]. Akesson et al. [16] reported that naturally occurring polysaccharides such as gums, agars, and algin serve as thickening agents in water-based applications. Guargum based polysaccharides can effectively reduce the percentage of spray droplets $\leq$ 150 $\mu$m [17] by altering the viscoelastic properties of the spray solution [18]. Extensional viscosity allows spray droplets to resist liquid stretching. Shear viscosity is the level of viscosity at a given shear rate. As shear viscosity decreases, spray droplets become coarser [14]. Altering both extensional viscosity and shear viscosity can produce a coarser droplet with a higher volume median diameter (VMD) thereby reducing drift potential [14].

The combination of DCAs with certain formulations of glyphosate may result in decreased efficacy [13]. Jones et al. [15] reported the addition of two differing DCAs to glyphosate resulted in 19 and 50% less spray volume with droplets < 141 $\mu$m and 15 and 59% larger VMD of spray droplets, when compared to glyphosate alone. When application pressure was increased 1.5-fold, droplets size effects were found to be proportionally similar to those of the original spray droplets [15].

Two other subgroups of the polyacrylamides include the nonionic polyacrylamides and the anionic polyacrylamides [14]. Anionic polyacrylamides are characterized by a negative net charge and a higher molecular weight. The nonionic subgroup is characterized by a net neutral charge and a lower molecular weight [14]. Additional DCAs consist of suspended polyacrylamides in an oil surfactant which forms an emulsion when in a spray solution [19]. Invert emulsions consist of water suspended in the oil phase causing the invert concentrate to encapsulate the pesticide with the droplet also encapsulating water. Invert emulsions increase VMD and reduce the driftable fraction [20].

A deposition aid is defined as a material that improves the ability of pesticide sprays to deposit on targeted surfaces [21]. There are two primary methods to increase pesticide deposition:(1) increase the level of pesticide deposited directly on the crop, and (2) increase uniformity of pesticide deposition [14]. Farris [22] reported an increase in the number of droplets observed per cm$^2$ of the target surface when a deposition aid was added to the spray mix. Increasing the level of a pesticide reaching the target surface has two primary benefits including increased application efficacy as well as reduced off target movement [14]. Richards et al. [23] observed that several deposition aids had no impact on the VMD or the driftable fraction of spray droplets during pesticide application.

The advent of herbicide-tolerant crops have increased herbicide application timing options across a growing number of hectares [24]. Troublesome weeds such as Palmer amaranth (*Amaranthus palmeri*) and waterhemp (*Amaranthus tuberculatus*) have an extended

germination period which poses a challenge POST herbicide application timing [25]. During late season herbicide applications the spray must pass through the crop canopy to reach the target weed species, where the crop canopy can intercept the spray and reduce spray deposition on target plants [24]. Understanding the spray canopy penetration of different spray solutions is important to optimize pesticide applications. The primary reason applicators use DRTs is to reduce off-target movement of herbicides during application. However, the impact of deposition aids on canopy penetration is not well understood [24]. Therefore, this research was initiated to determine the effects of varying deposition aids on herbicide combinations that may be used in cotton *Gossympium hirsutum* (L.) and soybean (*Glycine max* (L.) Merr.).

## 2. Materials and Methods

### 2.1. Spray Deposition Study

Experiments were conducted in 2014 and 2015 at the R. R. Foil Plant Science Research Center at Starkville, MS, USA, and the Black Belt Branch Experiment Station in Brooksville, MS, USA. Cotton and soybean were planted on conventionally tilled beds spaced 96 cm apart at 111,150 seeds ha$^{-1}$ and planting depth of 2 cm. Cotton (Deltapine 1321 B2RF, Monsanto Company, St. Louis, MO, USA) was planted on 8th May in 2014 and 3rd May in 2015 at Starkville, and 20th May in 2014 and 17th May in 2015 at Brooksville. Cotton seeds were treated with Acceleron N (Thiamethoxam + Pyraclostrobin + Ipconazole + Abamectin) (Monsanto Company, St. Louis, MO, USA). Soybean (Asgrow 5332, Asgrow Seed Co LLC, Creve Coeur, MO, USA) were planted at 333,450 seeds ha$^{-1}$ with 3.8 cm depth on 29 May in 2014 and 27 May in 2015 at both locations. Both crops were maintained until plants reached V3 (soybean) and first bloom (cotton). Soybean and cotton plants were ~61 cm height during treatment applications. Pesticide applications were made using a Bowman Mudmaster (Bowman Manufacturing, Newport, Arkansas) equipped with AIXR 110015 spray nozzles (TeeJet Technologies Spraying Systems Co., Glendale Heights, IL, USA) calibrated to deliver 138 L ha$^{-1}$ at 386 kPa. Applications were made 46 cm above the crop canopy with wind speed at or below 8 km hr$^{-1}$. All other management factors of the crop including fertility, weed, and insect pest management were applied and managed based on Mississippi State University extension recommendations (Mississippi State University Extension).

Separate, yet similar experiments were conducted in cotton and soybean. Both experiments utilized a factorial arrangement of treatments within a complete randomized design with all experiments containing four replications. Herbicide deposition aids were evaluated for each crop. Herbicide treatments for the cotton field study (Table 1) consisted of: glyphosate + dicamba (MON 76832, Monsanto Company, St. Louis, MI, USA), glyphosate + 2, 4-D (Enlist Duo$^{TM}$ with Colex D technology, Dow AgroSciences LLC, Indianapolis, IN, USA), glufosinate (Liberty® 280 SL, Bayer CropScience, Durham, NC, USA) + dicamba (Xtendimax® with VaporGrip® technology, Monsanto Company, St. Louis, MI, USA), glyphosate + dicamba (MON 76832) + acetochlor (Warrant®, Monsanto Company, St. Louis, MI, USA), and glyphosate + dicamba (MON 76832) + *S*-metolachlor (Dual Magnum, Syngenta Crop Protection, Greensboro, NC, USA).

Soybean herbicide treatments (Table 1) consisted of glufosinate + dicamba, glyphosate + 2, 4-D (Enlist Duo) + glufosinate, glyphosate (Roundup Powermax, Monsanto Company, St. Louis, MI, USA) + fomesafen (Flexstar®, Syngenta Crop Protection, Greensboro, NC, USA), glufosinate + fomesafen, and glyphosate + dicamba (MON 76832) + fomesafen.

All herbicide treatments were applied alone or in combination with HM 9679A (oil) at 1% *v/v*, HM 1428 (polymer) at 0.5% *v/v*, or HM 9733 (guargum) at 30 g/38 L of water. All deposition aids were provided by Helena Chemical Company. A fluorescent red tracer dye (Cole-Parmer, Vernon Hills, IL, USA) was added to each treatment solution at 0.2% *v/v*.



**Table 1.** Herbicide tank-mixtures tested in the cotton and soybean spray deposition study.

| Crop | Herbicide Treatments | Products Used | Rate (kg ae/ai ha$^{-1}$) |
|---|---|---|---|
| Cotton | glyphosate + dicamba | MON76832 | 1.1 + 0.6 |
| | glyphosate + 2,4-D | Enlist Duo$^{TM}$ | 0.9 + 0.9 |
| | glufosinate + dicamba | Liberty$^®$ 280 SL + Xtendimax$^®$ | 0.6 + 0.6 |
| | glyphosate + dicamba + acetochlor | MON76832 + Warrant$^®$ | 1.1 + 0.6 + 1.3 |
| | glyphosate + dicamba + *S*-metolachlor | MON76832 + Dual Magnum | 1.1 + 0.6 + 1.38 |
| Soybean | glufosinate + dicamba | Liberty$^®$ 280 SL + Xtendimax$^®$ | 0.6 + 0.6 |
| | glyphosate + 2,4-D + glufosinate | Enlist Duo$^{TM}$ + Liberty$^®$ 280 SL | 0.9 + 0.9 + 0.6 |
| | glyphosate + fomesafen | Roundup Powermax + Flexstar$^®$ | 1.1 + 0.26 |
| | glufosinate + fomesafen | Liberty$^®$ 280 SL + Flexstar$^®$ | 0.6 + 0.26 |
| | glyphosate + dicamba fomesafen | MON76832 + Flexstar$^®$ | 1.1 + 0.6 + 0.26 |

Metal stands were constructed and utilized to evaluate deposition into the crop canopy. Stands measured 61 cm in height with each stand made up of square tubing serving as the main beam. A horizontal cardholder was located on each of the four sides, and cardholders were spaced equidistantly apart on the vertical axis with the first position being located at the top of the canopy, second position being 15 cm downward from the first position, third position being 30 cm downward from the first position, and the fourth position being 46 cm downward from the first position. Each cardholder was positioned in a spiral manner down the main beam in an attempt to capture deposition throughout the crop canopy. Stands were placed in rows two and three of the crop row in each plot. The stand in row two was at the one end of the plot and the stand in row three was at the opposite end of the plot. Stands were placed in the row in a manner to which the lowest position was perpendicular to the row on one stand and parallel to the crop row on the other stand.

Mylar cards (Grafix, Maple Heights, OH, USA) (100 cm$^2$) were placed at each position on the stand and held securely using a small paper clip (Figure 1). All mylar cards were removed from each card holder using a fresh pair of latex gloves. Prior to the application process, mylar cards were placed on all stands and stand positions. Once the application had been made, the spray solution was allowed to dry for 90 seconds after which cards were collected. Cards were placed in a pre-labeled plastic bag and immediately placed in a dark container to avoid tracer photodegradation.

### 2.2. Fluorometric Analysis

Once all applications were completed, mylar cards were placed in cold storage and subsequently shipped to the University of Nebraska-Lincoln Pesticide Application Technology Laboratory located at the West Central Research and Extension Center in North Platte, NE. Spray deposition cards were evaluated with fluorimetry analysis [26,27]. Mylar cards were washed using 40 ml of a 1:1 solution of distilled water and 91% isopropyl alcohol using a bottle top type dispenser (Model 6000-BTR LabSciences, Inc., Reno, NV, USA). With the tracer completely suspended, a 2 mL aliquot was transferred to glass cuvette and analyzed using a Trilogy$^®$ fluorimeter with a rhodamine module (Turner Designs, Sunnyvale, CA, USA) at 24 C. Spray deposition was estimated in relative fluorescence units (RFU). Additionally, random samples were further diluted to bring RAW fluorescence unit readings within a range known response of the calibrated fluorometry system [28].

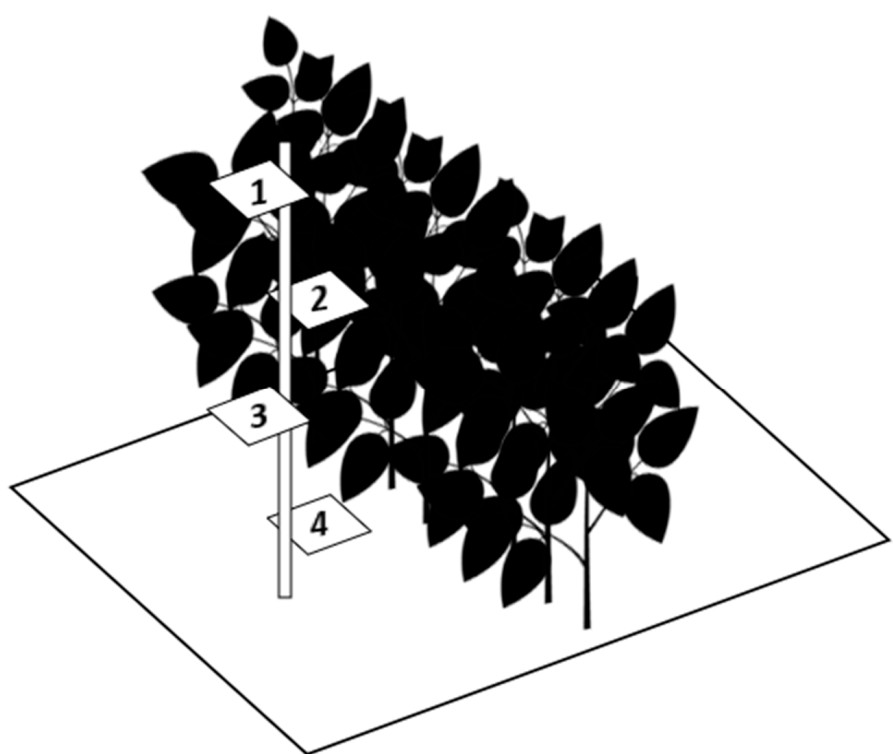

**Figure 1.** Spray canopy deposition study diagram. Mylar cards were positioned on top canopy—61 cm (1), 15 cm downward (2), 30 cm downward (3), and 46 cm downward (4).

*2.3. Droplet Size Distribution Analysis*

After herbicides and deposition aids were mixed and prior to the application being made, 180 mL samples were collected from each treatment and placed in a dark container and stored at 3 °C. Samples were delivered to the University of Nebraska Pesticide Application Technology Laboratory for droplet size analysis. Droplet size from each product was determined using laser diffraction. The wind tunnel operates at a constant wind speed of 24 km hr$^{-1}$ with wind being directed in a laminar manner. Creech et al. [2] and Henry et al. [29] provide in depth details regarding wind tunnel operation. All pesticides and deposition aids were applied using a single nozzle (AIXR 110015) calibrated to deliver 138 L ha$^{-1}$ at 386 kPa. A Sympatec Helos/Vario KR (Sympatec Inc., Clausthal, Germany) laser diffraction system was utilized to collect three separate droplet size measurements for each spray solution which served as three replications.

*2.4. Statistical Analysis*

Spray deposition data from cotton and soybean were analyzed separately using SAS 9.4 (SAS Institute, Cary, NC 27513, USA). Data were analyzed using the PROC GLIMMIX procedure with means separated using Fischer's protected LSD of $\alpha = 0.10$. Droplet size data were analyzed using SAS 9.4. using the PROC GLIMMIX procedure and means separated using Fischer's protected LSD of $\alpha = 0.05$.

## 3. Results

*3.1. Cotton Field Study*

Spray deposition data were pooled across years. When herbicides were pooled, deposition aids did not influence spray deposition on cotton ($p = 0.82$) (Table 2). As expected, Mylar card position within cotton canopy influenced spray deposition ($p < 0.0001$). When treatment solutions were pooled, Mylar cards positioned on top canopy (position 1) had 1972.8 RFU (relative fluorescence units), whereas cards positioned on the bottom of the canopy had 913.0 RFU (Table 3). When herbicide solutions were evaluated individu-

ally, the use of deposition aids influenced spray deposition for glyphosate + dicamba + *S*-metolachlor ($p$ = 0.03), glyphosate + dicamba + acetochlor ($p$ = 0.0003), and glyphosate + 2,4-D ($p$ = 0.0058). For applications of glyphosate + dicamba + *S*-metolachlor, the addition of polymer (1640.6 RFU) and guargum based (1407.3 RFU) deposition aids increased overall spray deposition compared to applications without deposition aids (807.4 RFU). The addition of oil (916.2 RFU) did not improve spray deposition compared to herbicide alone applications (Table 4). For glyphosate + dicamba + acetochlor applications, the inclusion of polymer (1179.3 RFU) and oil based (668.2 RFU) deposition aids decreased the spray deposition on cotton when compared to herbicide alone applications (2238.9 RFU) (Table 4). The oil deposition aid improved 2,4-D + glyphosate spray deposition (1389.7 RFU) on cotton compared to the guargam (798.3 RFU), the polymer (695.4 RFU), and the herbicide alone application (931.7 RFU) (Table 4).

**Table 2.** Analysis of variance and associated *p*-values for deposition aids and canopy levels influencing spray deposition on cotton. Data were pooled over years.

| Effect | Degrees of Freedom | F-Value | *p*-Value |
|---|---|---|---|
| Deposition aid | 3 | 0.31 | 0.8195 |
| Position within canopy | 3 | 18.79 | <0.0001 |
| Deposition aid × canopy | 9 | 0.13 | 0.9988 |

**Table 3.** Spray deposition [a] for different cotton canopy levels. Data were pooled over years. Means within a column followed by the same letter are not significantly different ($\alpha \leq 0.10$).

| Canopy Position | Spray Deposition (RFU) [b] |
|---|---|
| 1 (Top canopy) | 1972.82 A |
| 2 (15 cm downward) | 1327.57 B |
| 3 (30 cm downward) | 1219.94 B |
| 4 (46 cm downward) | 913.02 C |

[a] Data were pooled over herbicide solutions; [b] Relative fluorescence units.

**Table 4.** Cotton spray deposition [a] of herbicide solutions tank-mixed with different deposition aids. Data were pooled over years. Means within a column followed by the same letter are not significantly different ($\alpha \leq 0.10$).

| Deposition Aid | Spray Deposition (RFU) [b] | | |
|---|---|---|---|
| | Glyphosate + Dicamba + *S*-Metolachlor | Glyphosate + Dicamba + Acetochlor | Glyphosate + 2,4-D |
| Guargam | 1407.27 AB | 2036.51 A | 798.29 B |
| No deposition aid | 807.43 C | 2238.85 A | 931.65 B |
| Oil | 916.19 BC | 668.24 B | 1389.74 A |
| Polymer | 1640.55 A | 1179.33 B | 695.39 B |

[a] Data were pooled over canopy levels. [b] Relative fluorescence units.

### 3.2. Droplet Size Distributions of Herbicide Solutions Used in Cotton

An interaction between herbicide and deposition aid impacted all droplet parameters evaluated including: $DV_{0.1}$, $DV_{0.5}$, $DV_{0.9}$, relative span, and % droplets < 150 μm (Table 5). $DV_{0.1}$, $DV_{0.5}$, and $DV_{0.9}$ refer to the droplet size in which 10, 50, and 90% of the spray volume is of a lesser droplet diameter, respectively. Relative span is a dimensionless response variable that provides a measure of the spray droplet distribution homogeneity.

**Table 5.** Analysis of variance and associated *p*-values for droplet size parameters of herbicide solutions and deposition aids applied to cotton.

| Effect | D.F. [A] | $D_{V0.1}$ [B] | $D_{V0.5}$ [C] | $D_{V0.9}$ [D] | RS [E] | <150 μm [F] |
|---|---|---|---|---|---|---|
| Herbicide | 4 | ≤0.0001 | ≤0.0001 | ≤0.0001 | ≤0.0001 | ≤0.0001 |
| Deposition Aid | 3 | ≤0.0001 | ≤0.0001 | ≤0.0001 | ≤0.0001 | ≤0.0001 |
| Herbicide × Deposition Aid | 12 | ≤0.0001 | ≤0.0001 | ≤0.0001 | ≤0.0001 | ≤0.0001 |

[A] Degrees of Freedom. [B] Droplet diameter which 10% of the spray volume are contained in droplets of smaller diameter. [C] Droplet diameter which 50% of the spray volume are contained in droplets of smaller diameter. [D] Droplet diameter which 90% of the spray volume are contained in droplets of smaller diameter. [E] Relative Span. [F] Percentage of droplets < 150 μm.

Generally, the addition of a deposition aid increased $DV_{0.1}$, $DV_{0.5}$, and $DV_{0.9}$ values. However, inclusion of an oil deposition aid with glyphosate + 2, 4-D and glyphosate + dicamba + *S*-metolachlor resulted in reduced $DV_{0.1}$ and $DV_{0.9}$ values respectively. Inclusion of an oil deposition aid with glyphosate + dicamba with or without acetochlor or *S*-metolachlor resulted in similar $DV_{0.5}$ as to when these products were applied with no deposition aid. Applications of glyphosate + 2,4-D with or without an oil deposition aid resulted in similar $DV_{0.5}$ values. Applications of glyphosate + dicamba alone or with acetochlor resulted in similar $DV_{0.9}$ values when applied with no deposition aid or when applied with an oil. Noting the previous exception, inclusion of guargum deposition aid produced the largest $DV_{0.1}$, $DV_{0.5}$, and $DV_{0.9}$ values followed by polymer and oil deposition aids compared to where no deposition aid was utilized.

Relative span was impacted by herbicide and deposition aid interaction and there were no apparent trends associated with relative span due to treatments (Table 6). Lower relative span indicates a more homogenous spray distribution. The inclusion of an oil deposition aid reduced the relative span of all herbicide(s) except glyphosate + 2,4-D. Similar relative span values were observed when an oil or polymer deposition aid was added to glyphosate + dicamba and dicamba + glufosinate. The addition of a polymer deposition aid increased the relative span of glyphosate + 2,4-D, glyphosate + dicamba + acetochlor or *S*-metolachlor compared to relative span values observed when no deposition aid or an oil deposition aid were included with these herbicides. The addition of a guargum deposition aid to glyphosate + dicamba or dicamba + glufosinate resulted in smaller relative span values than where no deposition aid was utilized or where oil or polymer deposition aids were utilized. However, adding a guargum deposition aid to glyphosate + 2,4-D or glyphosate + dicamba + acetochlor or *S*-metolachlor resulted in relative span values greater than when no deposition aid or an oil deposition aid was utilized. Generally, the greatest relative span values were observed when a polymer deposition aid was used.

Herbicide and deposition aid interaction impacted the % of droplets less than 150 μm in size (Table 5). The % of droplets less than 150 μm ranged from 0.6–16% depending on the product. The addition of a deposition aid, regardless of type, decreased the % of droplets less than 150 μm, with the one exception occurring when glyphosate + 2,4-D was combined with an oil deposition aid which produced more droplets < 150 μm when compared to glyphosate + 2,4-D without a deposition aid. The greatest reduction in % of droplets less than 150 μm with all herbicide combinations was attained when a guargum deposition aid was utilized. With all herbicide(s), the polymer deposition aid produced fewer droplets less than 150 μm in diameter when compared to treatments that received no deposition aid or treatments that utilized an oil deposition aid.

**Table 6.** Effect of cotton herbicide combinations and deposition aids on spray $D_{V0.5}$, $D_{V0.1}$, $D_{V0.9}$, relative span, and <150 μm. Means within a column followed by the same letter are not significantly different ($\alpha \leq 0.05$).

| Herbicide Combinations | Deposition Aid | $D_{V0.1}$ [A] | $D_{V0.5}$ [B] | $D_{V0.9}$ [C] | RS [D] | <150 μm [E] |
|---|---|---|---|---|---|---|
| | | μm | | | | % |
| glyphosate + dicamba | Alone | 154 M | 337 M | 566 IJ | 1.22 E | 9.37 C |
| | Oil | 174 J | 346 L | 564 IJK | 1.13 I | 6.33 G |
| | Polymer | 240 F | 532 F | 848 E | 1.14 GHI | 2.90 M |
| | Guargum | 304 C | 660 C | 1031 B | 1.1 J | 1.36 P |
| glyphosate + 2,4-D | Alone | 194 H | 357 K | 551 K | 1.00 L | 4.35 I |
| | Oil | 187 I | 356 K | 565 IJ | 1.06 K | 5.08 H |
| | Polymer | 215 G | 457 I | 781 G | 1.24 CDE | 3.3 L |
| | Guargum | 310 B | 675 B | 1081 A | 1.14 GHI | 1.13 Q |
| glufosinate + dicamba | Alone | 121 P | 289 P | 519 L | 1.38 A | 15.58 A |
| | Oil | 146 O | 326 N | 553 JK | 1.25 BCD | 10.57 B |
| | Polymer | 217 G | 476 H | 802 F | 1.23 DE | 3.61 K |
| | Guargum | 365 A | 728 A | 1093 A | 1.00 L | 0.58 R |
| glyphosate + dicamba + acetochlor | Alone | 164 K | 348 L | 568 I | 1.16 G | 7.75 F |
| | Oil | 173 J | 345 L | 564 IJK | 1.13 HI | 6.46 G |
| | Polymer | 215 G | 484 G | 808 F | 1.23 DE | 4.00 J |
| | Guargum | 267 D | 599 D | 985 C | 1.20 F | 1.99 O |
| glyphosate + dicamba + *S*-metolachlor | Alone | 150 N | 316 O | 513 L | 1.15 GH | 9.98 B |
| | Oil | 158 L | 313 O | 495 M | 1.08 JK | 8.49 E |
| | Polymer | 197 H | 447 J | 763 H | 1.26 B | 4.94 H |
| | Guargum | 251 E | 569 E | 966 D | 1.27 BC | 2.53 N |

[A] Droplet diameter which 10% of the spray volume are contained in droplets of smaller diameter. [B] Droplet diameter which 50% of the spray volume are contained in droplets of smaller diameter. [C] Droplet diameter which 90% of the spray volume are contained in droplets of smaller diameter. [D] Relative Span. [E] Percentage of droplets < 150 μm.



### 3.3. Soybean Field Study

Spray deposition data were pooled across years. When herbicides were pooled, deposition aids did not influence spray deposition on soybean ($p = 0.34$) (Table 7). For pooled treatment solutions, Mylar cards positioned on top canopy (position 1) had 2657.4 RFU, whereas cards positioned on the bottom of the canopy had 779.7 RFU (Table 8). When herbicide solutions were evaluated individually, the use of deposition aids influenced spray deposition for glyphosate + 2,4-D + glufosinate ($p = 0.081$), and glufosinate + fomesafen ($p = 0.10$). The addition of deposition aids did not influence overall spray deposition for glufosinate + dicamba ($p = 0.79$), glyphosate + fomesafen ($p = 0.81$), and glyphosate + dicamba + fomesafen ($p = 0.72$) tank solutions. For applications of glyphosate + 2,4-D + glufosinate, the addition of oil (1307.7 RFU) and guargum based (1222.8 RFU) deposition aids decreased overall spray deposition compared to applications without deposition aids (1886.7 RFU) and with polymer deposition aid (1857.0 RFU) (Table 9). For glufosinate + fomesafen applications, the addition of polymer (1748.7 RFU) and oil (1535.0 RFU) deposition aids increased overall spray deposition when compared to applications without deposition aids (984.9 RFU). The addition of guargum deposition aid (1288.3 RFU) did not improve spray deposition compared to the herbicide alone solution (Table 9).

**Table 7.** Analysis of variance and associated *p*-values for deposition aids and canopy levels influencing spray deposition on soybean. Data were pooled over years.

| Effect | Degrees of Freedom | F-Value | *p*-Value |
|---|---|---|---|
| Deposition aid | 3 | 1.12 | 0.3402 |
| Position within canopy | 3 | 55.58 | <0.0001 |
| Deposition aid × canopy | 9 | 0.90 | 0.5249 |

**Table 8.** Overall spray deposition [a] for different soybean canopy levels. Data were pooled over years. A Means within a column followed by the same letter are not significantly different ($\alpha \leq 0.10$).

| Canopy Position | Spray Deposition (RFU) [b] |
|---|---|
| 1 (Top canopy) | 2657.38 A |
| 2 (15 cm downward) | 2038.85 B |
| 3 (30 cm downward) | 951.15 C |
| 4 (46 cm downward) | 779.70 C |

[a] Data were pooled over herbicide solutions. [b] Relative fluorescence units.

**Table 9.** Soybean spray deposition [a] of herbicide solutions tank-mixed with different deposition aids. Data were pooled over years. Means within a column followed by the same letter are not significantly different ($\alpha \leq 0.10$).

| Deposition Aid | Spray Deposition (RFU) [b] | |
|---|---|---|
| | Glyphosate + 2,4-D+ | Glufosinate + Fomesafen |
| | Glufosinate | |
| Guargam | 1222.79 B | 1288.34 AB |
| No deposition aid | 1886.72 A | 984.92 B |
| Oil | 1307.67 B | 1535.04 A |
| Polymer | 1856.98 A | 1748.67 A |

[a] Data were pooled over canopy levels. [b] Relative fluorescence units.

### 3.4. Droplet Size Distributions of Herbicide Solutions Used in Soybean

An herbicide by deposition aid interaction was present for spray droplet size distribution (Table 10). $D_{V0.1}$ ranged from 120 μm to 365 μm. With the exception of adding an oil deposition aid to glufosinate + 2,4-D or fomesafen, adding an oil, polymer or guargum deposition aid increased $D_{V0.1}$ (Table 11). Regardless of herbicide(s), the addition of polymer

deposition aid resulted in greater $D_{V0.1}$ than those observed following addition of an oil deposition aid or no deposition aid. Furthermore, addition of guargum to all herbicide(s) resulted in the greatest $D_{V0.1}$.

**Table 10.** Analysis of variance and associated *p*-Values for droplet size parameters of herbicide solutions and deposition aids applied to soybean.

| Effect | D.F. [A] | $D_{V0.5}$ [B] | $D_{V0.1}$ [C] | $D_{V0.9}$ [D] | RS [E] | <150 μm [F] |
|---|---|---|---|---|---|---|
| Herbicide | 4 | ≤0.0001 | ≤0.0001 | ≤0.0001 | ≤0.0001 | ≤0.0001 |
| Deposition Aid | 3 | ≤0.0001 | ≤0.0001 | ≤0.0001 | ≤0.0001 | ≤0.0001 |
| Herbicide × Deposition Aid | 12 | ≤0.0001 | ≤0.0001 | ≤0.0001 | ≤0.0001 | ≤0.0001 |

[A] Degrees of Freedom. [B] Droplet diameter which 10% of the spray volume are contained in droplets of smaller diameter. [C] Droplet diameter which 50% of the spray volume are contained in droplets of smaller diameter. [D] Droplet diameter which 90% of the spray volume are contained in droplets of smaller diameter. [E] Relative Span. [F] Percentage of droplets < 150 μm.

**Table 11.** Effect of soybean herbicide combinations and deposition aids on spray $D_{V0.5}$, $D_{V0.1}$, $D_{V0.9}$, relative span, and < 150 μm. Means within a column followed by the same letter are not significantly different ($\alpha \leq 0.05$).

| Herbicide Combinations | Deposition Aid | $DV_{0.1}$ [A] | $DV_{0.5}$ [B] | $DV_{0.9}$ [C] | R.S. [D] | <150 μm [E] |
|---|---|---|---|---|---|---|
| | | μm | | | | % |
| glufosinate + dicamba | Alone | 121 P | 289 O | 519 K | 1.38 A | 15.58 B |
| | Oil | 146 M | 326 K | 553 J | 1.25 D | 10.57 E |
| | Polymer | 217 G | 476 G | 802 F | 1.23 DE | 3.61 K |
| | Guargum | 365 A | 728 A | 1093 A | 1.00 L | 0.58 P |
| glufosinate + 2,4-D | Alone | 133 O | 310 L | 542 J | 1.32 B | 12.75 C |
| | Oil | 135 O | 304 MN | 500 LM | 1.20 EFG | 12.55 C |
| | Polymer | 179 J | 407 J | 700 I | 1.28 C | 6.35 H |
| | Guargum | 352 B | 705 B | 1068 B | 1.01 L | 0.66 P |
| glufosinate + fomesafen | Alone | 120 P | 281 P | 489 M | 1.31 B | 16.29 A |
| | Oil | 121 P | 286 OP | 495 LM | 1.31 B | 15.81 B |
| | Polymer | 195 I | 432 I | 723 H | 1.22 DE | 5.06 I |
| | Guargum | 334 C | 689 C | 1072 B | 1.07 K | 0.92 O |
| glyphosate + fomesafen | Alone | 158 L | 326 K | 543 J | 1.18 GH | 8.44 F |
| | Oil | 163 K | 321 K | 524 K | 1.12 J | 7.47 G |
| | Polymer | 228 F | 497 F | 823 E | 1.20 EFG | 2.97 L |
| | Guargum | 292 D | 647 D | 1025 C | 1.13 IJ | 1.50 N |
| glyphosate + dicamba + fomesafen | Alone | 139 N | 300 N | 496 LM | 1.19 FG | 12.00 D |
| | Oil | 147 M | 310 LM | 505 L | 1.16 HI | 10.56 E |
| | Polymer | 202 H | 443 H | 739 G | 1.21 EF | 4.53 J |
| | Guargum | 264 E | 588 E | 982 D | 1.22 DE | 2.09 M |

[A] Droplet diameter which 10% of the spray volume are contained in droplets of smaller diameter. [B] Droplet diameter which 50% of the spray volume are contained in droplets of smaller diameter. [C] Droplet diameter which 90% of the spray volume are contained in droplets of smaller diameter. [D] Relative Span. [E] Percentage of droplets < 150 μm.

$D_{V0.5}$ values ranged from 281 μm to 728 μm. The addition of an oil deposition aid to 2,4-D + glufosinate reduced $D_{V0.5}$ values. With the exception of adding an oil deposition aid to glyphosate + fomesafen or glufosinate + fomesafen; addition of all other deposition aids to herbicides increased $D_{V0.5}$ values. Adding an oil deposition aid to glyphosate + fomesafen or glufosinate + fomesafen resulted in similar $D_{V0.5}$ as when no deposition aid was utilized. Addition of a guargum deposition aid resulted in the greatest $D_{V0.5}$ values, regardless of herbicide. Inclusion of a polymer deposition aid resulted in lower $D_{V0.5}$ values compared to when a guargum was utilized. However, addition of a polymer

deposition aid increased $D_{V0.5}$ values compared to values obtained when no deposition aid or an oil deposition aid was utilized.

$D_{V0.9}$ values were maximized when a guargum deposition aid was utilized, regardless of herbicide(s). Inclusion of an oil deposition aid with glufosinate + fomesafen or glyphosate + dicamba resulted in similar $D_{V0.9}$ values to when no deposition aid was utilized. The use of a polymer deposition aid decreased $D_{V0.9}$ values compared to those obtained when guargum was used but increased $D_{V0.9}$ values compared to where an oil deposition aid was used.

The effect of deposition aid and herbicide on relative span varied depending on product combination. The addition of an oil deposition aid decreased the relative span of droplets from all herbicides except glufosinate + fomesafen compared to when no deposition aid was used. Use of a polymer deposition aid decreased relative span of droplets when applying glufosinate + dicamba, 2,4-D, or fomesafen compared to when no deposition aid was used. However, the addition of polymer deposition aid to glyphosate + fomesafen or glyphosate + dicamba resulted in similar relative span of droplets as when no deposition aid or guargum deposition aid, respectively, were included. Inclusion of guargum with all herbicide(s) except glyphosate + dicamba + fomesafen decreased relative span of spray droplets compared to when no deposition aid was used as well as when an oil or polymer deposition aid was utilized.

The greatest percent of spray droplets < 150 μm was observed when no deposition aid was utilized, regardless of herbicide. The inclusion of a guargum, polymer, and oil deposition aid resulted in the least to greatest percent of droplets < 150 μm produced. When no deposition aid was utilized, the greatest percentage of droplets < 150 μm were produced, regardless of herbicide.

## 4. Discussion

In both field studies, a variety of results were produced. Treatment combinations were evaluated for droplet size in the wind tunnel to better understand field results. Creech et al. indicated that nozzle design, operating pressure, herbicide solution, nozzle orifice size, and carrier volume, in order of greatest impact to least, influence droplet size [2]. General herbicide application guidelines indicate that reduced droplet sizes are necessary for contact herbicides to maximize spray coverage and efficacy, whereas systemic herbicide efficacy is less sensitive to changes in droplet size and can be sprayed with coarser droplets [30]. After reviewing several studies in the literature, Knoche [13] indicated that in general herbicide performance increased as droplet size decreased. Interestingly, the author noted that the performance of systemic herbicides increased more consistently with decreasing droplet size when compared with herbicides with a contact mode of action. This highlights that spray droplet size plays an important role in herbicide efficacy and should be tailored to the herbicide being used and the targeted weed species [13,31].

In the cotton field study, two opposite ends of the droplet spectrum were represented in the highest RFU values. Three of the five treatments produced $D_{V0.1}$ < 200 μm. Glyphosate + dicamba + oil deposition aid, glufosinate + dicamba + oil deposition aid, and glyphosate + dicamba + acetochlor with no deposition aids resulted in $D_{V0.1}$ < 200 μm. Generally, smaller droplets increased the level of spray deposition. Application of dicamba + glufosinate resulted in the greatest relative span of droplets and the highest percentage of droplets < 150 μm. These findings would agree with findings from Spillman [32] and Forster et al. [33] who found that smaller droplets with a lower terminal velocity resulted in greater leaf retention. Glyphosate + dicamba + fomesafen produced the smallest spray droplet size. Lake [34] found that smaller droplets with less terminal velocity had greater leaf retention because they were less likely to bounce.

In cotton, glufosinate + dicamba applied with the guargum deposition aid produced the largest droplet sizes with the lowest relative span, as well as the lowest percentage of droplets ≤ 150 μm (Table 6). Relative fluorescence units observed with this treatment was not different of those produced from applications without a deposition aid. In all other

scenarios where RFU values were above 2200, all treatment combinations had a relative span between 1.13 and 1.25. Generally, when the relative span was on the lower end of this range, the treatment combination had a percentage of droplets $\leq 150\ \mu m$ above 6% and less than 11% (Table 6). This would suggest that variability in droplet size as well as the number of droplets produced $\leq 150\ \mu m$ can complement one another with respect to crop canopy penetration.

Differences associated in the level of deposition measured in RFUs at each position between canopy types (Tables 3 and 8) can potentially be explained through the management of each crop. Mepiquat chloride is applied to cotton as a plant growth regulator. Applications of mepiquat chloride reduce length between internodes and plant height by reducing gibberellic acid in plant tissues [35,36]. Reduced gibberellic acid causes cell walls to stiffen, reduced elongation, and slower division of cells [37–39]. In both years of this study, mepiquat chloride was applied at First bloom to cotton. These applications can change the plant architecture by reducing distance between nodes, and if additional applications are not warranted, node length can begin to expand. Moreover, seeding rates vary greatly between the two crop types. Furthermore, the leaf area index (LAI) differs between the two crop types. The optimum LAI for soybean in a subtropical environment is between 3.6 (indeterminate) and 4.5 (determinate) at first flower. The leaf area index of soybean has been observed to reach 6.0 to 6.5 indeterminate and determinate cultivars [40]. Yield potential of cotton has been shown to maximize at a LAI of 5 [41]. The greatest cotton deposition was measured at the top of the crop canopy (position 1). No differences between 15 cm and 30 cm (position 2 and 3) downward from the top were detected. Deposition measured at the bottom of the cotton canopy (45 cm downward) was reduced when compared to positions 1 and 2 (Table 3). Deposition measured in the soybean canopy was significantly greater at the top position when compared to positions (3 and 4) (Table 8). Deposition was not reduced from the top position to 15 cm downward. A reduction in the level of deposition was obtained when moving from 15 cm to 30 cm. However, there were no differences in deposition between the position located 30 cm downward from the top and the position located 45 cm downward from the top (Table 8). Data from both studies support findings from Wolf and Bretthauer [42] who also observed differing levels of deposition inside a crop canopy. Results from this study may have varied if wind speed was greater at the time of application, or if deposition could have been measured on an actual leaf surface instead of a mylar card.

Creech et al. [24] reported that the use drift reducing adjuvants did not influence spray canopy penetration for herbicide applications on corn and soybean. The authors also reported an interaction between spray carrier volume, nozzle design, and the use of adjuvants influencing spray deposition on corn canopy. The addition of adjuvants increased spray deposition on corn for applications at 94 L ha$^{-1}$, whereas deposition was decreased when adjuvants were added for applications at 187 L ha$^{-1}$. Creech et al. also observed a general trend in increased herbicide spray deposition as adjuvants were used [24]. Results from this study corroborates remarks from Creech et al. [24] regarding the complexity of pesticide applications and the need for tailoring nozzle selection, adjuvants, active ingredients, and application rates for every pesticide application scenario.

## 5. Conclusions

Although deposition aids influenced spray deposition on cotton and soybean for some herbicide combinations, the type of deposition aid and the use of deposition aids should be determined on a case-by-case scenario. In several instances, total deposition decreased when a deposition aid was combined with the herbicide treatment. In both laboratory studies, the addition of a deposition aid affected all droplet size parameters. However, at no point in time in either field study was the utilization of a deposition aid observed to have a positive impact on the level of canopy penetration. Recommending the use of deposition aids with products in these situations could have a negative impact on a grower's profit margin. However, under different environmental conditions the utility of

deposition aids could prove to have a positive impact on the level of canopy penetration observed in the field. Therefore, additional research is necessary to better understand how deposition aids can influence herbicide applications and consequent spray deposition and canopy penetration. Further studies are necessary to better understand the interactions between droplet size, spray carrier volume, solution physicochemical properties, and canopy structure influencing spray deposition and efficacy during pesticide applications. Spray deposition strategies must account for all droplet dynamics and the relationship between spray deposition and biological efficacy.

**Author Contributions:** Conceptualization, C.A.S., T.R.B., J.T.I., D.B.R., D.M.D.; formal analysis, C.A.S., T.R.B., B.C.V., J.T.I., G.R.K., D.M.D.; writing—review & editing, C.A.S., T.R.B., B.C.V., J.T.I., D.B.R., A.C., G.R.K., D.M.D., investigation, C.A.S., T.R.B., J.T.I., D.B.R.; project administration, D.M.D., resources, D.M.D., supervision, D.M.D., methodology D.M.D., G.R.K. All authors have read and agreed to the published version of the manuscript.

**Funding:** This research received no external funding.

**Data Availability Statement:** The datasets generated and/or analyzed during the current study are available from the corresponding author on reasonable request.

**Acknowledgments:** This study was partially supported by Bayer CropScience. No conflicts of interest have been declared.

**Conflicts of Interest:** The authors declare no conflict of interest.

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
