# Peer review of "Effect of Deposition Aids Tank-Mixed with Herbicides on Cotton and Soybean Canopy Deposition and Spray Droplet Parameters"

_agronomy, doi:10.3390/agronomy11020278_

Round 1

Reviewer 1 Report

Reviewer 1

This research paper was initiated to determine the effects of varying deposition aids on herbicide combinations that may be used in cotton [Gossympium hirsutum (L.)] and soybean [Glycine max (L.) Merr.]. Because the impact of deposition aids on canopy deposition is not well understood. In the research work was commonly used herbicides and applied on soybean and cotton in combination with deposition aids (oil, polymer, and guargum) and was found for the polymer-based deposition aid increased spray deposition on cotton for applications of glyphosate+dicamba+S-metolachlor resulting in 1640.6 RFU (relative fluorescence units). However, the same deposition aid decreased spray deposition on cotton for applications of glyphosate+dicamba+acetochlor (1179.3 RFU). I have some questions about this paper.    

  1. No picture of the experiment is given in the research work. I urge the authors to attach at least two or three pictures relating to the material and methods of research work (e.g. a picture of the test polygon, picture of the sprayer, implementation plan of the experiment…).
  2. the authors in the results describe the Droplet size and velocity distribution measured in RFU (relative fluorescence units). Whether you also performed an analysis of the number of droplets in the research work, e.g. on an area of 1 cm2, this parameter is also important factors in accuracy and retention of pesticide applications on velocity distribution.
  3. Also you dont describe future work (please include in your research paper for example section 5. Conclusions and future work), nowhere in the article did I spot you describing about the future work … e.g. advanced application techniques for herbicide application and comparison with other research works.

In my point of view, need to be imroved and attach at least some pictures in the research work. I suggest that you also reconsider to including broader interpretation about future work, this will get your research work for application technique to more attention in terms of environmental protection. Depending on your revisions i will reconsider whether it will be your paper suitable for publication in journal Agronomy.

Reviewer 2 Report

The topic of this study is very interesting. In this article, the authors evaluated the effects of different deposition aids tank-mixed with herbicides a) on canopy deposition in cotton and soybean crops and b) on spray droplets parameters. This article presents information and experimental results that are worth publishing. I provide below several suggestions that, if the authors decide to implement into the paper, the paper will improved.

Comments

Tittle of the article: The title of article should be revised including the term “spray droplet parameters”

Abstract: The abstract should be revised.  The authors should be present data about the effects of deposition aids on spray droplets parameters.

Introduction: This section is well written. The authors should a short paragraph about the importance of proper application to achieve high herbicide efficacy.

Line 58, 73, 76 the publication year should be deleted. The authors should follow the instructions for authors.

Material and methods: This section should be improved. In the first paragraph of this section the authors should add more information about the time (and cotton + soybean growth stage) of herbicides application. Usually, in no GMO crops the post-emergence herbicides applied at early growth stages in cotton and soybean. So, the authors should add information about the growth stage of crop during the herbicides application. Moreover, the authors should explain why it is important the study of herbicides deposition into the crop canopy in these crops. It is common in USA, the herbicides to apply when the plants (cotton or soybean) reached 61 cm height?

Table 1 and 2 should be merged into one table

Table 4, 5, 6, and 7 should be merged into one table

Table 12 and 13 should be merged into on table

Results: This section is well written.

Discussion: This section should be revised. The authors should add more references regarding the deposition of herbicides into the canopy, while it is important to mention why it is important this? Moreover, the authors should a few references regarding the importance of droplet size for herbicides application and its efficacy.

References: The references should be corrected according to instructions for authors.

Round 2

Reviewer 2 Report

The authors applied the suggested changes to the manuscript during the reviewing process and improved the manuscript. Thus, this article can be accepted for publication on Agronomy.